# Atypical Staphylococcal Septic Arthritis in a Native Hip: A Case Report and Review

**DOI:** 10.3390/pathogens12030408

**Published:** 2023-03-03

**Authors:** Ira Glassman, Kevin H. Nguyen, Michelle Booth, Marine Minasyan, Abby Cappadona, Vishwanath Venketaraman

**Affiliations:** 1College of Osteopathic Medicine of the Pacific, Western University of Health Sciences, Pomona, CA 91766, USA; 2WesternU Health Patient Care Center, Western University of Health Sciences, Pomona, CA 91766, USA

**Keywords:** *Staphylococcus aureus*, septic arthritis, diagnosis, diabetes, tobacco, mPCR

## Abstract

Septic arthritis is a synovial fluid and joint tissue infection with significant morbidity and mortality risk if not diagnosed and treated promptly. The most common pathogen to cause septic arthritis is *Staphylococcus aureus*, a Gram-positive bacterium. Although diagnostic criteria are in place to guide the diagnosis of staphylococcal septic arthritis, there is a lack of adequate sensitivity and specificity. Some patients present with atypical findings which make it difficult to diagnose and treat in time. In this paper, we present the case of a patient with an atypical presentation of recalcitrant staphylococcal septic arthritis in a native hip complicated by uncontrolled diabetes mellitus and tobacco usage. We review current literature on diagnosing *S. aureus* septic arthritis, novel diagnostic technique performance to guide future research and assist clinical suspicion, and current *S. aureus* vaccine development for at-risk patients.

## 1. Introduction

*Staphylococcus aureus* is a Gram-positive bacterium capable of antimicrobial evasion, invasive infections, sepsis, and death. In 2019, *S. aureus* was the only bacterial pathogen associated with more than 1.1 million global deaths, approximately 10,000 of which were from bone and joint infections [1]. The pathogen is more broadly referred to in community and health-care facility infections as methicillin-susceptible *S. aureus* (MSSA) or methicillin-resistant *S. aureus* (MRSA). While MRSA hospital-acquired and community acquired infections have declined slowly during 2013–2016, hospital-acquired MSSA infections have not changed significantly, and community acquired MSSA infections have slightly increased (3.9% per year) from 2012 to 2017 in the United States [2].

*S. aureus* is the most common cause of septic arthritis [3]. Septic arthritis is an infection of synovial fluid and joint tissues which causes significant morbidity and mortality when not diagnosed and treated promptly. Timely diagnosis is crucial to minimizing osseous destruction and necrosis and preventing long-term disability [4,5]. Even with antibiotic usage, there is a 7–15% mortality rate for in-hospital septic arthritis and 1/3 of patients experience morbidity [3]. A nationwide, multi-centric retrospective study in France of septic arthritis by Richebé et al. found that surgery was required in 48.3% of patients and during follow-up, 28.3% had serious complications and 9.2% died, with *S. aureus* associated with higher mortality [6].

Diagnosis of septic arthritis follows a typical pattern of clinical suspicion by symptom presentation, analysis of serum markers, imaging, and synovial fluid collection. Common acute presentations include fever, edema, erythema, functional limitation, tenderness to palpation, and if chronic, a draining sinus communicating with the joint. More than 50% of patients with septic arthritis have a history of joint swelling, joint pain, and fever [7]. It often presents as monoarticular in medium-large joints, as demonstrated by a retrospective study of 248 cases of native joint infection by Kennedy et al. with 92.3% of cases being monoarticular [3,8]. 

Currently, no imaging finding is pathognomonic for septic arthritis in adults. X-ray imaging may be useful in revealing widened joint spaces, but subchondral bone changes are a late finding, with subchondral bone changes presenting late in the course of infection. Ultrasonography is useful in guiding needle aspiration to improve yield, especially in cases with a dry arthrocentesis finding, as well as visualizing joint effusion [3]. Magnetic resonance imaging (MRI) is the gold standard for assessing osteomyelitis and soft tissue infections, preferably with and without contrast [7]. While MRI can demonstrate soft tissue changes to guide clinical suspicion, there are limitations to its use ranging from incompatible pacemakers and metallic hardware and reduced availability in rural and remote regions [9]. In 2019, a study found that in Minnesota, US, a predominantly rural state, 39 out of 87 counties did not have an in-hospital MRI machine [10].

Laboratory serum markers are helpful but not diagnostic. Hariharan et al. found that serum ESR and CRP levels are each more than 90% sensitive for septic arthritis when low cutoffs are used (ESR > 15 mm/h and CRP > 20 mg/L), which can prove useful in ruling out septic arthritis cases [11]. Kocher et al. include ESR greater than or equal to 40 mm/hr as a factor in their 4-variable model including inability to bear weight, fever, and peripheral leukocytosis, demonstrating 99.6% predictive accuracy in patients with all four symptoms [12]. 

Analysis of synovial fluid obtained via arthrocentesis is useful in the differentiation of septic arthritis from other forms of arthritis and to determine the causative pathogen. Synovial fluid analysis should include Gram staining, aerobic and anaerobic cultures, and white blood cell count with differential [7]. Despite synovial fluid culture from arthrocentesis being the gold standard for septic arthritis confirmation, methods of direct sampling and standard cultures are negative in 10–30% of cases despite infection being present [13]. Balato et al. found that culture negative infections have been reported to range from 16.7% to 78.4% [14]. A nine-year retrospective analysis at a single center in Portugal by Cipriano et al., with predominantly *S. aureus* septic arthritis, noted that only 50.5% of arthrocentesis cultures had positive growth in synovial fluid [15]. Another retrospective study by Daynes et al. analyzing 183 cases of native joint infections found that joint fluid cultures were positive in 55% of cases; and of these cases, blood cultures yielding the same pathogen were found in 54% [16].

Clinical suspicion also requires knowledge of risk factors which predispose patients to infection. Known risk factors for septic arthritis include: diabetes mellitus, HIV and immunosuppression, intravenous drug use, prosthetic joints, age older than 80, smoking, rheumatoid arthritis and osteoarthritis, sexual activity, recent surgery, skin infection, and iatrogenic causes [7]. Cipriano et al. also noted diabetes as the most commonly reported comorbidity with 20.6% of patients having diabetes [15]. A five-year single-center retrospective study in New Zealand of native joint septic arthritis by McBride et al. found that 35% of patients used tobacco and 24% of patients had diabetes mellitus [17]. These risk factors can also be associated with increased severity of disease, where Hunter et al. found that a history of diabetes increased risk for recalcitrant septic arthritis [18].

In this paper, we focused our attention on a patient who presented with recalcitrant MSSA septic arthritis in a native hip joint with atypical diagnostic findings, uncontrolled diabetes mellitus, and a history of tobacco usage. We aim to present this patient’s case and review current literature on the current state of diagnostic methods, novel methods in identifying *S. aureus* septic arthritis, and the status of *S. aureus* vaccine development for at-risk patients like ours.

## 2. Materials and Methods

To find studies for this article on the current diagnostic methods of septic arthritis, a series of steps were performed. This included collecting data on keywords, inclusion, and exclusion criteria. Information was obtained using PubMed and NCBI databases within the period from 2000 to 2023. Search results included terms such as: “*Staphylococcus aureus*”, “MSSA”, “septic arthritis”, “diagnostic”, “mPCR”, and “diabetes”. Attention was paid in each section to include articles that were relevant to *S. aureus* septic arthritis of native joints, diagnostic methods, medium-large joint infections, and diabetes comorbidity. Articles in other languages, as well as articles with abstracts but no full-texts available were excluded. Other exclusion criteria included non-relevance to *S. aureus* septic arthritis, small joints, prosthetic, periprosthetic, post-operative joint infections, and pediatric cases. The results yielded 507 articles, of which 21 were included based on the above criteria.

## 3. Case Presentation

### 3.1. Case History

Here, we report the case of a 47-year-old male presenting to the emergency department (ED) with right hip pain which arose suddenly after work and a past medical history of type 2 diabetes mellitus and tobacco usage. He presented with right lower extremity pain and limited range of motion without erythema, swelling, or fever. Multiple superficial injuries present on shins bilaterally, secondary to motorcycle and recreational use. X-ray imaging was normal, and the patient was discharged without blood tests. He was diagnosed with right hip pain, prescribed cyclobenzaprine for muscle spasm and ibuprofen for pain, and informed to follow up with his primary care physician (PCP).

One month later, the patient presented to his PCP who subsequently identified elevated white blood cell count (WBC, 21,000/L) and referred the patient to the ED. Vital signs were normal and blood tests showed elevated WBC (11,000/L), glucose (291 mg/dL), erythrocyte sedimentation rate (ESR, 104 mm/hr), and c-reactive protein (CRP, 21 mg/dL). Physical exam noted right lower extremity pain with internal rotation and external rotation, unable to perform a straight leg raise, diffusely tender to palpation over hip flexors, and no appreciable swelling or erythema. The patient was able to ambulate to bathroom but had discomfort bearing weight. Repeat X-ray of the hip showed normal anastomotic alignment without any fracture or destructive bone lesion but there were mild osteoarthritic changes. An arthrocentesis was attempted but no fluid could be aspirated. Subsequent MRI revealed extensive edema surrounding musculature and soft tissue of the right proximal femur. Repeat blood cultures sampled over 48 h revealed presence of *S. aureus* and antimicrobial susceptibility testing was performed, shown in Section 3.2. The patient was started on intravenous (IV) vancomycin. An incision and drainage (I&D) procedure was performed with drainage culture negative. A wound culture from adjacent tissue during the I&D was positive for the same strain of MSSA from the blood cultures. The patient was transitioned to IV cefazolin via peripherally inserted central catheter (PICC) line after I&D procedure. Repeat blood cultures 4- and 5-days post-I&D yielded no growth, and the patient was discharged.

One month later, the patient was admitted to a new hospital with worsened symptoms. His blood tests showed normal WBC (8490/L), elevated glucose (217 mg/dL), elevated ESR (135 mm/hr), and elevated CRP (4.4 mg/dL), with CRP and ESR values seen in Table 1. X-ray and three-dimensional CT (3D CT) revealed severe joint space narrowing at the right hip joint with associated erosion of the acetabulum, seen in Figure 1 and Figure 2. Computerized tomography (CT) scan of abdomen and pelvis with contrast revealed osseous erosion involving the right acetabulum and right proximal femur with adjacent 9.6 cm abscess. MRI of pelvis revealed loss of normal marrow signal and cortex erosive changes of right femoral head and neck, as well as right acetabulum, and multifocal right groin muscular myositis. MRI of left spine revealed multilevel degenerative changes with moderate to severe bilateral neural foraminal narrowing at L5-S1. He was diagnosed with MSSA septic arthritis, a 9.6 cm abscess lateral to right greater trochanter, osteomyelitis, and bacteremia. I&D, adjacent fluid collection, resection of femoral head (Girdlestone procedure), and fabrication and insertion of resorbable antibiotic beads impregnated with vancomycin and gentamicin was performed. The patient was discharged to a nursing care facility with continued IV cefazolin via PICC, instructed to inject 20 mL (2000 mg total) into the vein every 8 h for 38 days, which is a commonly used antibiotic for MSSA in the United States. He was referred to infectious disease for monitoring and orthopedic surgery for hip replacement surgery. He presented to the PCP for follow up with improved symptoms and new diabetes management one week later.

The PICC line was removed 6 weeks after surgery and the patient was re-evaluated. Incision site had healed with no drainage, erythema, or swelling surrounding, and the patient was improving. Blood cultures continued to be positive for MSSA, and he was placed on oral doxycycline 100 mg BID for two months, which is utilized for resistant strains of *S. aureus* in the United States. The patient has since concluded their antibiotic series, controlled their blood sugar and bacteremia, and received a full hip replacement without complications.

### 3.2. Antimicrobial Susceptibility Testing

The Kirby-Bauer disk diffusion method was used to determine the antimicrobial susceptibility of the *S. aureus* strain according to Clinical and Laboratory Standards Institute (CLSI) guidelines [19]. The strain was susceptible to amoxicillin-clavulanic acid, clindamycin, erythromycin, gentamicin, levofloxacin, linezolid, oxacillin, quinapristin/dalfopristin, tetracycline, sulfamethoxazole-trimethoprim, and vancomycin, and resistant to penicillin.

## 4. Discussion

We report a case of a patient who suffered significant morbidity from *S. aureus* septic arthritis infection due to a delay in diagnosis. Our patient had several diagnostic pitfalls that led to the delay in their care and subsequent morbidity. With atypical clinical presentation, normal imaging, dry-arthrocentesis and culture-negative synovial fluid from I&D, the patients only signs were elevated inflammatory markers and bacteremia until the infection had spread too far. *S. aureus* commonly grows easily in culture and has features that allow for rapid detection such as its characteristic gold-colored colonies from the yellow pigment staphyloxanthin, Gram-positive grape-like cluster morphology, positive catalase test, and ability to coagulate blood and plasma, indicated by its positive coagulase test [20]. Despite this, *S. aureus* remains a difficult pathogen to culture when involved in bone and joint infections. This may be due to its ability to grow intracellularly within host osteoblasts, which is accomplished through modifying their genetic expression to survive in low numbers for a prolonged time as small-colony-variants [21]. Another explanation may be the ability of *S. aureus* to form biofilms, adhering as a group in bones and joints for protection. Trouillet-Assant et al., demonstrated that recurrent *S. aureus* bone and joint infection yielded a reduced inflammatory response, survived longer intracellularly in host osteoblasts, was less cytotoxic, and formed more biofilms [22]. Novel detection methods to improve diagnostic yield require attention to improve health outcomes in patients like ours.

ESR and CRP are inflammatory markers which can assist clinical suspicion of disease processes but are not diagnostic, as they are elevated in numerous conditions. Zhao et al. revealed that procalcitonin has a higher specificity than CRP when using a cutoff of 0.5 ng/mL or greater and may deserve attention as a marker for septic arthritis [23]. Akdoğan et al. found that the sensitivity, specificity, and accuracy of synovial fluid procalcitonin was 60%, 100%, and 78%, respectively [24]. They also investigated the use of human neutrophil peptides 1-3 (HNP 1-3) with values of 86%, 87%, and 87%, respectively [24]. Lastly, they concluded that when used in conjunction, the two markers in synovial fluid accurately diagnosed 97% of cases [24].

Diagnostic yield in patients with septic arthritis may be improved with the use of blood culture bottles for synovial fluid in addition to conventional culture onto solid media. While data is limited for native infections, studies have been published to assess the strength of these sampling methods. Kuo et al. sampled synovial fluid inoculated into blood culture bottles from 77 periprosthetic joint infection patients. They used both direct and routine matrix-assisted laser desorption ionization time-of-flight mass spectrometry (MALDI-TOF MS) and found that both were able to detect 80% of specimens, 47% of which were *S. aureus* [25]. Biendo et al. compared yield from blood culture bottles with the yield from synovial fluid inoculated in blood culture bottles, utilizing Gene Xpert MRSA/SA BC and confirmed results with MALDI-TOF MS. They successfully identified three out of three *S. aureus* synovial fluid infections and found no difference between blood culture bottles and biological fluid inoculated blood culture bottles [26]. In addition to the use of blood culture bottles for synovial fluid, bone biopsy is considered the gold standard for osteomyelitis and may improve yield for pathogens such as *S. aureus* which can form biofilms and live intracellularly [27].

Advancements in PCR methods have been investigated to improve diagnostic yield in patients with septic arthritis. Coiffier et al. employed broad-range PCR of 16s ribosomal DNA (16S rDNA PCR), in which nearly half of the patients had *S. aureus* septic arthritis and found that the 16s rDNA PCR had lower yield than direct examination, blood cultures, and synovial culture [28]. Labetoulle et al. tested a *S. aureus* specific PCR, the MRSA/SA ELITe MGB PCR Assay (Elitech), against 16s rDNA PCR. Elitech detects species specific sequences within the *ldh1* gene of *S. aureus* for identification, *mecA* and *mecC* genes for methicillin resistance, and an exogenous external control termed CPE. They found that the 16S rDNA PCR correctly identified presence of *S. aureus* in 23 out of 40 (56.5%) of samples whereas Elitech correctly identified 39 out of 40 (97.5%) of samples [29]. 

Sigmund et al. compared automated multiplex PCR (mPCR), utilizing the mPCR Unyvero i60 ITI (Unyvero, Curetis) according to manufacturer’s protocol, to synovial fluid culture. They found that the performance was not significantly different but that Unyvero mPCR improved detection time, with results within five hours [30]. Tarabichi et al. conducted a retrospective, multicenter study at two tertiary centers in the United States and Germany and found that synovial fluid culture median time to positivity for MSSA is 1.95 days [31]. Important to note that this technique does not reveal susceptibility and thus still requires culture. Unyvero mPCR was also compared to synovial fluid culture by Morgenstern et al. who found its performance inferior [32]. In comparison, Saeed et al. evaluated the BioFire Joint Infection Panel (BioFire) mPCR for synovial fluid against routine culture in a multicentre retrospective study in the UK and Ireland. They found that BioFire mPCR had an overall sensitivity of 91.6% and specificity of 93%. More specific to *S. aureus*, BioFire mPCR correctly identified 41 samples of *S. aureus* whereas routine culture correctly identified 37 samples [33]. Shoenmakers et al. evaluated the performance of BioFire mPCR in 45 synovial fluid samples from patients who had clinical suspicion of septic arthritis, demonstrating a sensitivity of 83% and specificity of 100% in native septic arthritis. Interestingly, they found a lower sensitivity of 73% for patients with suspected late acute prosthetic joint infection and 30% for those with suspected late acute prosthetic joint infection [34]. Based on current literature, it appears the BioFire mPCR may serve as an improvement over traditional synovial fluid culture and Unyvero mPCR, especially in native joint infections, but larger studies are needed to evaluate their relative strengths. 

Palmer et al. trialed PCR-electrospray ionization-time-of-flight mass spectrometry (molecular diagnostics [MDx]) against synovial fluid culturing and found that MDx identified bacteria in 50% of suspected septic arthritis cases, all of which were *S. aureus*, whereas culture detected only 16% [35]. MDx was found to be more sensitive as well, confirmed by species-specific 16s rRNA fluorescence in situ hybridization (FISH) [35]. This method can provide bacterial identification and antibiotic sensitivity prediction within 6 h, but still warrants further study as 50% of patients with clinical suspicion for septic arthritis were negative by MDx [35].

Blood cultures should be strongly considered and used to add to clinical suspicion and improve upon culture yield, especially in bacterial pathogens such as *S. aureus* which can spread to bone and joints by hematogenous routes. New research has been focused on improving detection methods serologically to limit invasive techniques and improve detection time in diagnosing septic arthritis. Sulovari et al. has designed an assay which permits the detection of immunoglobulin G (IgG) response to species-specific antigens to identify if an infection is present. Despite the culture negative prevalence of *S. aureus*, the team has had success in identifying *S. aureus* infection [13]. Nishitani et al. conducted multivariate analysis of IgG titers which found that no antigen titer was a consistent predictor of *S. aureus* infection but identified iron-regulated surface determinant protein B (isdB) antigen which was found in patients who were more likely to die from *S. aureus* infection [36]. IsdB and alpha hemolysin were identified as the two best single antigens in predicting ongoing *S. aureus* infection but require future research [36].

*S. aureus* remains a challenging pathogen to diagnose in septic arthritis, with modern methods failing to reach the performance necessary to identify it in time to reduce morbidity and mortality. Patients with risk factors such as diabetes and tobacco usage are at increased risk of infection and severity [15,17,18]. In addition to diagnostic method improvements, preventative research is also needed to mitigate morbidity associated with patients like ours. A major challenge in the development of a vaccine targeting *S. aureus* is the difficulty in translating results from animal models, specifically *murine*, to humans [37]. Vaccine development has focused on recombinant proteins, protein glycoconjugation, novel bioconjugation, extracellular vesicles, whole cell and live-attenuated cells, nucleic acids, as well as adjuvants to improve efficacy of these novel vaccine methods [38]. The most significant recent vaccine failure was Pfizer’s SA4Ag, specifically tested on patients undergoing spinal surgery. The vaccine was composed of recombinant proteins conjugated to a detoxified form of diphtheria toxin. Although the vaccine was found to be highly immunogenic in *murine* models, it was unable to reduce incidence of *S. aureus* infections 90- and 180-days post-surgery [38]. Preclinical results of vaccines show efficacy in *murine* models but fail to trigger a broad enough immune response when initiated in humans. As such, there is a need for better translational models, such as organoids [38]. 

Teymournejad et al. hypothesized that the failure to transition from *murine* models to humans stems from human exposure to *S. aureus* throughout life. They found that *S. aureus*-sensitized mice were unprotected by *S. aureus* skin and soft tissue infection (SSTI) following alpha-hemolysin-targeted vaccination. They found that prior exposure to SSTI did not inhibit vaccine-specific antibody response but did inhibit vaccine-specific T cell response and alpha-hemolysin-targeted vaccine efficacy. They also found that utilizing a Th1/Th17 stimulating adjuvant, CAF01, was able to restore vaccine efficacy in *the S. aureus*-exposed mice [39]. As all vaccine trials have been focused on stimulating humoral immunity, their results demonstrate the importance of generating cell-mediated immunity as an immunoadjuvant. 

In a novel approach, Pan et al. utilized *Lactobacillus reuteri* WXD171-IsdB, engineered as a delivery vessel displaying *S. aureus* antigen IsdB, which can tolerate the gastrointestinal environment to function as an oral vaccine with less potential risk than attenuated bacterial vectors such as diphtheria toxin. They found that *L. reuteri* WXD171-IsdB had a strong protective effect against *S. aureus* pulmonary infection, skin and soft tissue infection, systemic infection, and elicited enhanced humoral and importantly, cell-mediated immunity in *murine* models [40]. This novel method may serve to improve the immune response in humans and requires future research as a vaccine delivery system.

## 5. Conclusions

Diagnostic methods for identifying *S. aureus* septic arthritis promptly to reduce associated morbidity and mortality from infection are not optimal and require advancement in research and development. Clinical presentation may not always be reflective of infection and imaging is often a late finding, as evidenced by our patient. MRI continues to be the gold standard in imaging but has limitations in select populations and regional availability. Synovial fluid culture by arthrocentesis remains the gold standard for confirmation of pathogens however, cases of dry arthrocentesis and culture negative samples make reliability difficult. I&D procedures can improve upon yield but are invasive and share the potential for culture negative samples. Blood samples should be utilized in guiding clinical suspicion and improving culture sensitivity, as common septic arthritis pathogens such as *S. aureus* seed hematogenously in joints. Inflammatory markers such as ESR and CRP are helpful but not diagnostic. The utilization of species-specific PCR (Elitech) and novel mPCR (BioFire) demonstrate promising improvement in the detection of bone and joint infections over traditional synovial fluid culture, but larger studies are needed to evaluate their strengths. Developing preventative measures for high-risk patients, such as vaccination, remains a challenge, with clinical trials failing to reach adequate immune responses in humans. Focusing on cell-mediated immune responses may represent an important new frontier for *S. aureus* vaccine and immunoadjuvant development, but more research is needed to transition these *murine* results to human models.

## Figures and Tables

**Figure 1 pathogens-12-00408-f001:**
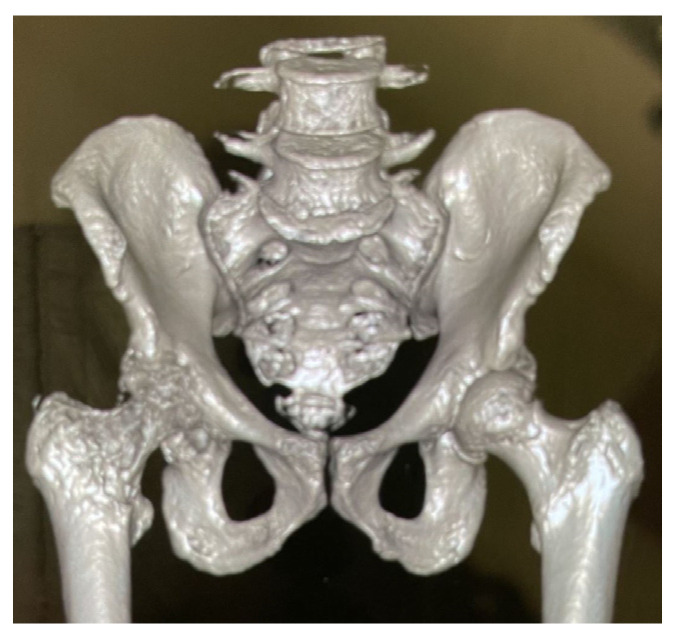
3D CT image of hip taken before Girdlestone procedure, 67 days after symptoms began.

**Figure 2 pathogens-12-00408-f002:**
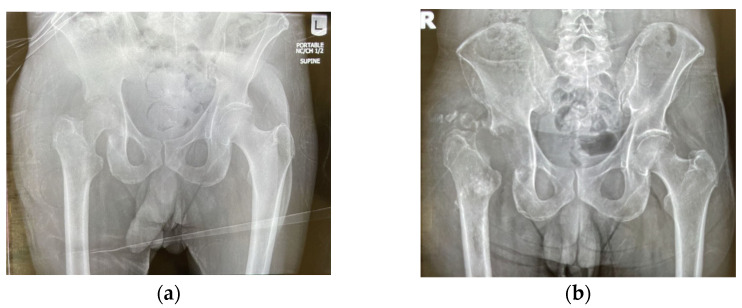
(**a**) X-ray of hip taken before Girdlestone procedure, 67 days after symptoms began; (**b**) X-ray of hip taken after Girdlestone procedure, 68 days after symptoms began.

**Table 1 pathogens-12-00408-t001:** CRP and ESR values from 5 days to 90 days since initial symptoms.

Days since Initial Symptoms	CRP (mg/dL)	ESR (mm/hr)
5 d (Pre-I&D)	21	104
35 d (Post-I&D/Pre-Girdlestone)	4.4	135
82 d (Post-Girdlestone)	2	47
90 d (Post-Girdlestone)	4	26

## Data Availability

Data sharing not applicable.

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
