# Peer review of "Atypical Staphylococcal Septic Arthritis in a Native Hip: A Case Report and Review"

_pathogens, 2023, doi:10.3390/pathogens12030408_

Round 1

Reviewer 1 Report (Previous Reviewer 3)

1. Line 53-54: Abbreviation form of  "(MRI) " should be added to "Magnetic resonance imaging".

2. Line 130-132. "polymerase chain rection (PCR) analysis" is scientifically primitive expression. Only "PCR" is enough.

3. Line 130-132. What PCR method/scheme was used to identify MSSA? Reference must be added to show PCR method.

4. For the MSSA isolate, was antimicrobial susceptibility tested? If so, its results with method must be added.

5. line 223, 246: S. aureus should be italicized. Check it throughout the manuscript.  

Author Response

Thank you for your feedback, we have included the above changes into the paper and believe it improves the quality of our paper.

Reviewer #1

Comments and Suggestions for Authors

  1. Line 53-54: Abbreviation form of  "(MRI) " should be added to "Magnetic resonance imaging".

Response: Corrected.

  1. Line 130-132. "polymerase chain rection (PCR) analysis" is scientifically primitive expression. Only "PCR" is enough.

 Response: Corrected.

  1. Line 130-132. What PCR method/scheme was used to identify MSSA? Reference must be added to show PCR method.

Response: While we do not have the original PCR method, the laboratory did confirm using traditional microbiological methods which have been added to the case presentation section.

  1. For the MSSA isolate, was antimicrobial susceptibility tested? If so, its results with method must be added.

 Response: Yes, we have added the antimicrobial susceptibility testing in the case presentation section.

  1. line 223, 246: S. aureus should be italicized. Check it throughout the manuscript.  

Response: Thank you, we have corrected this throughout the paper.

Reviewer 2 Report (Previous Reviewer 2)

The authors comprehensively expanded the introduction and added relevant epidemiological studies for reference. In order to give the reader an overview of current challenges in the diagnosis of SA, the authors provide additional references and further expand on the information about clinical features of the disease. The introduction in its current state meets all the criteria that are required in order to help the reader to understand the scientific and clinical background. The authors further added critical information to the methods section. In their case report, the authors provide additional clinical information and added a CT image of the patient’s hip that shows fulminant bone erosion. Importantly, the authors now state that they followed up on the patient’s state subsequent to surgery. The discussion section now connects literature data to the here presented case and allows the reader to integrate the patient’s history into a bigger picture.

Author Response

Reviewer #2

Comments and Suggestions for Authors

The authors comprehensively expanded the introduction and added relevant epidemiological studies for reference. In order to give the reader an overview of current challenges in the diagnosis of SA, the authors provide additional references and further expand on the information about clinical features of the disease. The introduction in its current state meets all the criteria that are required in order to help the reader to understand the scientific and clinical background. The authors further added critical information to the methods section. In their case report, the authors provide additional clinical information and added a CT image of the patient’s hip that shows fulminant bone erosion. Importantly, the authors now state that they followed up on the patient’s state subsequent to surgery. The discussion section now connects literature data to the here presented case and allows the reader to integrate the patient’s history into a bigger picture.

Response: Thank you for your kind words. We appreciate your guidance in evaluating our draft and we are glad the changes were sufficient.

Reviewer 3 Report (Previous Reviewer 1)

In this revised version of the manuscript, the authors have significantly modified the article in order to focus on the diagnostic challenges of S. aureus arthritis. They have removed all the elements about Corynebacterium tuberculostearicum, and the part about the research for vaccines against S. aureus is reduced to a paragraph of the discussion.

I still have major comments about this manuscript:

1) The manuscript was submitted to the section “Vaccines and Therapeutic Developments”. However, the part on the need for vaccines against S. aureus is very short and is not the main topic of the manuscript, so this section does not seem appropriate. This part should be expanded to fit the section.

2)     In my opinion, the review of literature about the diagnostic difficulties in S. aureus arthritis is not covered in sufficient detail. It does not provide much novelty for the reader. Moreover, the authors do not explain why in some cases there are diagnostic difficulties with S. aureus while it is usually an easy-to-culture bacterium. The bacterial mechanisms involved should be described (small colony variants, biofilm, intracellular bacteria).

3)    The authors present the performance of molecular assays and serological assays for the diagnosis of S. aureus arthritis but their data are not exhaustive. They presented 16S rDNA (“universal”) PCR and multiplex molecular assays, but they did not mention specific PCR targeting S. aureus while it has better performance than 16S PCR. For multiplex molecular assays, they did not specify the references of the reagents presented (manufacturer). Various multiplex PCR assays exist for bone and joint infections (Unyvero/Biofire) and they do not have the same performance, so the authors should clarify the results presented and compare the various assays in order not to generalize their performance.

Minor comments:

Line 44: The authors present the characteristics of septic arthritis in native joints, which are mainly acute. However, the “sinus communicating with the joint” only occur in chronic infections, so this should be clarified.

Lines 53-54: please introduce the abbreviation MRI “Magnetic resonance imaging (MRI)”

Line 112: did the patient have blood tests at the first visit (CRP, WBC)?

Lines 131-132: the sentence is not clear. What did the PCR confirm, the presence of S. aureus or the methicillin susceptibility of this S. aureus strain?

Line 152: please correct “gentamicin”

Line 193: another way to improve diagnostic yield in patients with septic arthritis is the use of blood culture bottles for synovial fluid or bone biopsies in addition to conventional culture onto solid media; it could be interesting to add it for readers.

Line 204: in septic arthritis, bacterial growth is usually rapid and the median time of 4.5 days presented here seems long. This result is based on a study including only 22 septic arthritis with a very low sensitivity of culture (10 patients with positive culture of synovial fluid). Therefore, it cannot be used as a “reference” for all septic arthritis and data from other publications should be added.

Author Response

Reviewer #3

Comments and Suggestions for Authors

In this revised version of the manuscript, the authors have significantly modified the article in order to focus on the diagnostic challenges of S. aureus arthritis. They have removed all the elements about Corynebacterium tuberculostearicum, and the part about the research for vaccines against S. aureus is reduced to a paragraph of the discussion.

I still have major comments about this manuscript:

  • The manuscript was submitted to the section “Vaccines and Therapeutic Developments”. However, the part on the need for vaccines against  aureusis very short and is not the main topic of the manuscript, so this section does not seem appropriate. This part should be expanded to fit the section.

Response: Thank you for your feedback. We agree that the section was condensed too briefly by our revision. We have expanded the vaccine section in the discussion to include evaluation of novel advancements in vaccine discovery for S. aureus, including mechanisms targeting the serological targets mentioned earlier in the discussion. We hope you like the additions.

  • In my opinion, the review of literature about the diagnostic difficulties in  aureusarthritis is not covered in sufficient detail. It does not provide much novelty for the reader. Moreover, the authors do not explain why in some cases there are diagnostic difficulties with S. aureuswhile it is usually an easy-to-culture bacterium. The bacterial mechanisms involved should be described (small colony variants, biofilm, intracellular bacteria).

Response: Thank you for your feedback. We have expanded upon the discussion of diagnostic difficulties to include bacterial mechanisms, a more thorough review of the mPCR differences mentioned below, as well as included more information on the utilization of blood culture bottles of synovial fluid culturing to improve diagnostic yield. We hope that these changes better embody the message we wish to convey.

  • The authors present the performance of molecular assays and serological assays for the diagnosis of  aureusarthritis but their data are not exhaustive. They presented 16S rDNA (“universal”) PCR and multiplex molecular assays, but they did not mention specific PCR targeting S. aureus while it has better performance than 16S PCR. For multiplex molecular assays, they did not specify the references of the reagents presented (manufacturer). Various multiplex PCR assays exist for bone and joint infections (Unyvero/Biofire) and they do not have the same performance, so the authors should clarify the results presented and compare the various assays in order not to generalize their performance. 

Response: Thank you for this. We have added species-specific PCR data to compare to the 165s rDNA to demonstrate the improved yield. We have also looked further into the different mPCR methods, i.e. BioFire, so as to not generalize only the Unyvero results. We believe that this greatly improves the quality of this section and appreciate your suggestions.

Minor comments:

Line 44: The authors present the characteristics of septic arthritis in native joints, which are mainly acute. However, the “sinus communicating with the joint” only occur in chronic infections, so this should be clarified.

Response: Thank you, we have clarified this now.

Lines 53-54: please introduce the abbreviation MRI “Magnetic resonance imaging (MRI)”

 Response: Corrected.

Line 112: did the patient have blood tests at the first visit (CRP, WBC)?

Response: The patient did not have blood tests at the first visit, we have added this.

Lines 131-132: the sentence is not clear. What did the PCR confirm, the presence of S. aureus or the methicillin susceptibility of this S. aureus strain?

Response: Thank you for this. We attempted to attain the original PCR methods, but we do have the original microbiological methods for determining the susceptibility. We have added the susceptibility methods to the case presentation section.

Line 152: please correct “gentamicin” 

 Response: Corrected.

Line 193: another way to improve diagnostic yield in patients with septic arthritis is the use of blood culture bottles for synovial fluid or bone biopsies in addition to conventional culture onto solid media; it could be interesting to add it for readers. 

Response: Thank you for this. We have added this information to the discussion section.

Line 204: in septic arthritis, bacterial growth is usually rapid and the median time of 4.5 days presented here seems long. This result is based on a study including only 22 septic arthritis with a very low sensitivity of culture (10 patients with positive culture of synovial fluid). Therefore, it cannot be used as a “reference” for all septic arthritis and data from other publications should be added.

Response: Thank you for catching this. We have removed this from the paper and added in a recent study highlighting culture time for MSSA.

Thank you again for your continued feedback on our paper. We appreciate your thoughts and advice and believe they greatly improve the quality of our paper. We want to present this case and information well.

Round 2

Reviewer 3 Report (Previous Reviewer 1)

In this second revised version of the manuscript, the authors have taken into account the reviewers' comments and modified the manuscript accordingly. The case presented is now detailed enough to understand the diagnostic difficulties encountered. The introduction provides all the elements necessary to understand the background. The discussion section has been expanded 1) to detail the mechanisms involved in difficult-to-diagnose S. aureus arthritis, 2) to present the performance of the different microbiological methods that may be useful in this context and 3) to give readers an overview of the state of research regarding S. aureus vaccines.

I just have a minor comment:

Lines 182-185: consider replacing “augmentin” by “amoxicillin+clavulanic acid” and “The growth of the strain was effectively inhibited by….” by “The strain was susceptible to …. and resistant to penicillin”

Author Response

Dear Reviewer,

Thanks for the constructive comments and positive feedback. We have revised the manuscript based on your comments below:

Lines 182-185: consider replacing “augmentin” by “amoxicillin+clavulanic acid” and “The growth of the strain was effectively inhibited by….” by “The strain was susceptible to …. and resistant to penicillin”

Based on your recommendation we have rephrased these sentences.

Once again, thank you. We look forward to publishing this important work.

This manuscript is a resubmission of an earlier submission. The following is a list of the peer review reports and author responses from that submission.

Round 1

Reviewer 1 Report

Please see the word file attached

Reviewer 2 Report

Septic Arthritis (SA) is a disabling and life-threatening disease. Much research is still needed to better understand its pathophysiology. Diagnosis of this disorder is still not optimal in the clinical setting and vaccines for the most important causative agent S. aureus are still lacking. Glassman et al. present a case of S. aures SA in a patient that later develops an invasive infection with C. tuberculostericum. The case report is well written and easy to understand. However, the case was not concluded and it would have been interesting to know the patients prospects of recovery. The authors continue to review the literature about challenges that must be overcome in order to properly manage SA in the clinical setting.

This article is of importance and every report about rare joint infections that are huge health burden has value to clinicians and researchers. The article is well written, however some aspects of it should be improved in order to help the reader better grasp the connection of the here presented case to the challenges for SA diagnosis, risk factor assessment, co-infection research and vaccine development.

Major concerns:

A. The methods section is very short. It should be mentioned how many articles were included/excluded for the literature review. It should be made clear that the methods for article selection were not subject to a selection bias.

B. Although the reader might grasp the context of the here presented case within the current state of challenges for SA management, the authors did not draw a line between the reviewed literature and the disease process for this particular patient. How do the challenges for SA diagnosis etc. apply to the case?

In the following, I provide a point-by-point review of the manuscript.

1. The abstract gives a good overview of the content, encourages the reader to continue and highlight the articles novelty, which is the presentation of a rare case involving coinciding C. tuberculostearicum infection.

2. The short introduction starts with S. aureus’ role in worldwide disease burden covering some important epidemiological background information specifically for MRSA. The section furthermore provides a short overview about S. aureus as the most important causative agent for SA.

a. Although the authors write about morbidities that are associated with SA, they should at least give one example that might elucidate the risks of disease progression (i.e. disability)

b. line 38: rather than citing a book (ref. 3) authors should focus on new research, reviews and epidemiological data

c. line 39: “The severity of infection is also exacerbated in patients who are more at-risk …” could the authors rephrase? This sentence seems pleonastic. Maybe just mention that there are risk factors that predispose for aggravated disease courses.

d. line 41: diagnosis criteria after Newman should at least be mentioned or cited: doi: 10.1136/ard.35.3.198

e. line 43: the authors mention that complication arises when a patient does not show typical symptoms of SA. This might not be entirely correct. Even when diagnosed, SA remains life-threatening and disabling. I would encourage the authors to add a fitting reference.

f. Shouldn’t the last paragraph have been written in another tense?

g. line 53: citation needed

3. The materials and methods section covers some criteria with which the review was written.

a. The authors should state how many articles were first selected and then used for review (after applying exclusion criteria).

4. The case is well presented in manner that is easily understandable even for a non-clinician. However, I am wondering why this case is now presented although the patient was still under care while this manuscript was written. Maybe I’m not familiar with the way case reports are published but shouldn’t the authors comment on the reasons for not waiting until the case concluded?   

a. line 81: where did this wound come from? Was MSSA identified in the blood OR in the wound OR both?

b. line 88: “His labs…” this reads like jargon and should be rephrased

c. lines 107: “1/4” what does this mean?

d. lines 90: it would have been interesting to see the CT images in contrast to the X-rays that did not show noticeable abnormalities.

5. In the section about diagnostic challenges the authors name some directives by which diagnosis of SA is usually performed. Furthermore, the authors mention some recent insights about how to overcome the challenges for diagnosis of SA, which adds to the value of this article. However, I find that the lack of context to the here presented case decreases the quality of this manuscript. What could have been done in their presented case that would have ensured a faster diagnosis and in turn would have prevented the perpetuation of the disease?

a. lines 121-122: “but subchondral bone changes are a late finding, …, with subchondral bone changes presenting late in the course of infection” This sentence should be rephrased to prevent redundancy.

b. ref. 6 is incomplete

 c. lines 128 – 131. The whole sentence is hard to comprehend as it mixes literature data with data from the here presented case. Please rephrase.

d. line 139: typo (full stop is missing)

6. The authors focus on risk factor as joint surgery and prosthetic joint replacement with some emphasis on the microbiology behind pathogen resilience (i.e. in biofilms). They continue to review some of the most important pre-existing diseases that predispose to SA and add some clues that might explain the interdependencies. However, most of the underlying mechanisms that are named here are prone to speculation and, if available, some laboratory work should be reviewed that investigated these mechanisms. The authors mainly cite from other reviews that did not generate primary data. Nonetheless this section is well written.

7. The authors next focus on the importance of the pathogen C. tuberculostearicum in joint infections. The section is interesting to read and highlight additional pit falls in the treatment of SA.

a. line 229: “et al.,…” typo (comma)

b. line 246: “BJI…” do you mean PJI?

c. line 247: “co-infection of C. tuberculostearicum 247 makes proper diagnosis more difficult” why?

d. line 252: “BJI infection” do you mean PJI? Also: redundancy (“I” stands for infection)

8. The authors continue with a well written outline of challenges concerning the development of an effective S. aureus vaccine. It is interesting to read, though a link to the here presented case should be made in order to help the reader to better follow the leitmotif of this article. The case report should be used as an example that is an epitome of the current state of diagnostic challenges for SA and further underscores the necessity for vaccine research as well as studies on atypical coinfection. The sections in this article in their current state seem to be only loosely connected and a reader might find it difficult to assemble the here reviewed data into a bigger picture.

9. Although the conclusion section is well written, the authors again draw no line to their specific case report.

Reviewer 3 Report

Major comments

1. Contents belonging to case report and review part should be more clearly separated. Otherwise, readers may confuse in understanding. Section 3. Case Presentation is OK. However, section 4. Diagnostic challenges includes Figure 1 and Table 1, and their explanation. In such case, readers understand that these are information of the present mentioned in section 3. If so, they should be moved to section 3. Check other descriptions to be moved to section 3.

2. In section 3, the method to identify MSSA and C. tuberculostearicum (C.t.) is not written. Identification methods must be written. Particularly, C.t. must have been identified by genetic methods. What evidence was obtained for identification of C.t. must be written. Reliability of species identification depends on the evidence.

3. Authors use "challenge" often in this manuscript. However, it seems to be somewhat exaggerated. Carefully use this word.

4. Sections 4, 5, 6, 8  are somewhat lengthy. It is better shorten, by deleting descriptions not related to the main topic directly. Section 8 may be deleted. Section 7 is missing. 

Minor comments

1. Authors write "gram-positive" in the manuscript. It must be corrected as "Gram-positive". Check and correct throughout the manusucript.

2. line 14: "there is a lack of..." this should be rephrased for better understanding by readers.

3. line 107:  "1/4" meaning of this is not clear, so add explanation.

4. line 202: Check whether "C'3" is correct or not.  

5. line 252: What is "BJI infection"?